# Assessing Stability of Crutch Users by Non-Contact Methods

**DOI:** 10.3390/ijerph18063001

**Published:** 2021-03-15

**Authors:** Achilles Vairis, Suzana Brown, Maurice Bess, Kyu Hyun Bae, Jonathan Boyack

**Affiliations:** 1Department of Mechanical Engineering, Hellenic Mediterranean University, Irákleion, 71410 Crete, Greece; vairis@hmu.gr; 2Department of Technology and Society, SUNY Korea, Incheon 21985, Korea; maurice.bess@stonybrook.edu (M.B.); KyuHun.Bae@stonybrook.edu (K.H.B.); jonathan.boyack@stonybrook.edu (J.B.)

**Keywords:** disability, assistive devices, stability, video analysis, acceleration, jerk, axillary crutches

## Abstract

Enhancing gait stability in people who use crutches is paramount for their health. With the significant difference in gait compared to users who do not require an assistive device, the use of standard gait analysis tools to measure movement for temporary crush users and physically disabled people proves to be more challenging. In this paper, a novel approach based on video analysis is proposed as non-contact low-cost solution to the more expensive alternative with the data collected from processed videos, two values are calculated: the Signal to Noise Ratio (SNR) of acceleration, and the Signal to Noise Ratio of the jerk (time derivative of acceleration), to assess the user’s stability while they walk with crutches. The adopted methodology has been tested on a total of 10 participants. Five are temporary users of assistive devices with one being a long-term user and the other four novice users, and five are disabled participants who use those assistive devices permanently. Preliminary results show differences between novice users, long-term users, and physically disabled users. The approach is promising and could improve the assessment of crutch user stability, allowing for the correction of gait for individuals while using an inexpensive non-contact setup and preventing unnecessary falls.

## 1. Introduction

Estimates from the World Health Organization (WHO) show that over one billion people need one or more assistive devices, with an expectation that these figures will double to beyond two billion by 2050 [1]. Among those assistive devices, the two most popular are walking and visual aids [2]. Walking aids accounted for 46.3% of all assistive devices [2] and were found to be more common in rural areas and among males while visual aids were more common in urban areas and among female users [2]. One explanation for the prevalence of visual and mobility aids is that both are relatively inexpensive and can be used effectively with little training or support while having the potential to make a big functional difference [2]. 

People who use assistive devices to walk are already unstable and in danger of further injury if experiencing a fall. The purpose of this paper is to evaluate one method with two parameters of measuring the stability of a person walking with the most common walking aid, a crutch. 

Stability is generally associated with gait [3]. Gait analysis is the assessment of the attributes or manner of walking done by observing a person as they walk in a straight line. According to some gait models, the symmetry and stability of gait constitute a gait appraisal model [4]. 

Typically, there are two ways to perform gait evaluation. The first is an evidence-based approach, while the second measures body kinematics, muscle activity, and body mechanics using sophisticated instrumentation. A number of studies into measuring gait have been conducted using high-end tools including force platforms, 3D cameras, and optical markers. These systems require installation in areas with sufficient space and are very expensive. A specialized technician is also required to properly operate the system [5]. The main interests in motion analysis currently revolve around detecting a specified activity or event (e.g., falls) [6]. Most research in this area uses linear acceleration and gyroscopes to detect falls by applying thresholds to velocities, accelerations, and angles [7]. It has been shown that inertial measurement systems (IMU) that use a combination of accelerometers and gyroscopes can reliably replace camera-based systems for clinical body motion and gait analyses [8].

In recent years, researchers have developed software that is able to detect and track human limbs using 135 key-points in a single image. OpenPose, one of the first real-time multi-person tracking systems, was developed by researchers at Carnegie Mellon University in 2014 [9]. It was released in the form of Python code, C++ implementation, and a Unity Plugin [10]. This multi-person 2D pose estimation method uses a specific nonparametric representation, Part Affinity Fields (PAFs), to learn to associate body parts with individuals in the image [10].

The OpenPose algorithm has shown to be capable of learning to associate human body parts with identifiable individuals within frames of a video [11], track and identify the action of doing bilateral squats [12] and be used to assess the movements of preschool children to diagnose gross motor action [13]. Since the creation of OpenPose, many pose-estimation systems with similar capabilities have been created. Multi-person pose estimation is expected to be applied to many fields, such as fitness training, pedestrian recognition, and military training [14]. To our knowledge, OpenPose has not been used to access stability dynamically and initial research regarding non-contact methods for remote assessment has only been explored recently [15].

Human joint stability is a parameter that is difficult to define accurately, or to quantify, as its assessment requires spatial and temporal measurements [16]. Studies of gait look into movement parameters of individual steps such as flight time or stance but further insight can be gained by studying the higher vector derivatives of displacement such as acceleration, which relates to the force experienced by the participants. Measurements of kinematic parameters of gait such as acceleration and jerk, the rate of change of acceleration, have been proposed to that effect, with abrupt accelerations or jerk related to poor control or stability [17]. For instance, jerk has been tested as the parameter of the sway in participants with Parkinson’s disease. Postural sway can be distinguished by calculating jerk on patients’ lower back [18]. Also, the jerk has been used as a parameter for measuring smoothness of movement. For example, a study found a correlation between jerk and proficiency of movement between experienced and novice dancers [19]. 

Fall prevention and stability is also a major concern in the field of rehabilitation and gait [20,21]. There are several studies [22,23] that use jerk as a measure of smoothness in movement. One model estimates the Cartesian kinematic jerk of the hips’ orientation during a three-dimensional movement by using a smartphone to collect gyroscopic data. The study finds that the Cartesian kinematic jerk grows in value as the analyzed motion becomes less fluent [22]. The other study proposes the movement fluency variables: hesitation, coordination and smoothness. The smoothness variable is based on the jerk of the total body center of mass. These parameters allow for the quantitative identification in changes to motion control due to age [23].

As the signals of such parameters tend to carry a lot of noise, differ even in people of the same age group and health status, and are affected by the measurement methods, additional processing is required to extract meaningful information from them. There are many studies in medical imaging systems that use Signal-to-Noise Ratio (SNR) for measuring the picture quality of images [24] in order to achieve improvements in penetration depth in medical ultrasound [25]. Increased SNR can also be used to allow imaging at higher frequencies and thereby increase spatial resolution without any loss of penetration [26]. SNR has been used extensively in signal processing [24,25,26], where there is noise interfering in the signal, and large values of SNR are considered good outcomes since it represents a strong signal with low noise. In other words, we calculate SNR as a measure of useful information to false information, as the acceleration of the participants in the present study incorporates noise. The use of SNR in our context is different and it has been calculated as the inverse of the coefficient of variation, and this is the first time that SNR has been used to assess the smoothness of movement. Because of this, the SNR that we calculate is dimensionless.

To our knowledge, there has been only one preliminary study assessing stability by measuring the acceleration of the people who walk with crutches [27], and that study looked at periodic patterns while using sensors and optical motion capturing of participants. In this paper, our objective is to eliminate expensive setups and simplify the measurements by using video analysis. The goal is to propose a simple, inexpensive, non-contact, remote evaluation of stability in crutch users.

The next section details methods and materials used in the experiment and measurements. 

## 2. Materials and Methods

This section discusses the experimental setup and methods applied to collect data. It also provides information about the participants and data collected from them.

### 2.1. Experimental Setup and Data

Video data collection was done at 30 frames per second using two 8-megapixel iPhone 6s set up on tripods. One camera recorded a side-view of participants at a distance far enough away to capture three to five steps when using swing-through gait. The other camera captured a front-view and was placed at an angle 90 degrees off from the side-view camera. This setup allowed for the capture of a stereoscopic view of the participants. A large checkerboard pattern was placed on the wall in view of both cameras to acquire the stereoscopic transformation matrix. This matrix was then used to obtain 3D positional data of objects within the recorded area. 

The two videos were manually synchronized by finding the frame where a major physical event occurred. This was typically a first step where either the foot of the subject or the crutch first left the ground or first contacted the ground that was visible in both videos. Since the framerates had been set to be equal, a certain number of frames was taken after that point to record the entire walk. This way the entire video sequence analyzed was synchronized and viewable in both videos and no extraneous material was run through the software.

The layout of the walkway is shown in Figure 1.

The 2D pixel location values of the joints were extracted from OpenPose and exported to the MATLAB software for 3D localization using stereo vision techniques. Furthermore, MATLAB extrapolated the 3D velocities, accelerations, and jerk from those positions.

After processing with OpenPose software, the visual data is presented in Figure 2.

To evaluate the movement of the participants the following time series values have been calculated from each video frame: acceleration (a) represented by the equation a = Δv/Δt, where v is the velocity and t is the time, and jerk (j), which is the second derivative of velocity, represented by the equation j = Δa/Δt. For each video frame, the instantaneous acceleration in space (i.e., in the x, y, and z coordinates) was calculated for the ankle of the weight-bearing leg for the whole run. The acceleration (a) has been calculated by combining the three acceleration components (α_x_, α_y_, α_z_ in the Cartesian coordinates) with the Equation (1).
(1)$$a=\sqrt{a_{x}^{2}+a_{y}^{2}+a_{z}^{2}}$$

Additionally, the data is processed to calculate Signal-to-Noise-Ratio (SNR) of acceleration and jerk. In practice, SNR is equivalent to the inverse of the coefficient of variation [28], or it is approximated by the ratio of the mean of all data (x_) divided by the standard deviation of all data (s) as shown in Equation (2), as variables are always non-negativeE ⁡ [ X 2 ] = σ 2 + μ 2 {\displaystyle \operatorname {E} \left[X^{2}\right] = \sigma ^{2}+\mu ^{2}}.
(2)SNR=xs

The SNR metric was selected in this study because while walking with crutches, the movement includes a large component of noise or abrupt changes superimposed stochastically to the signal. This reflects on the displacement data and the time derivatives of it which are velocity, acceleration and jerk. In order to capture all of the important information from the analysis of the video regarding gait and stability, the raw acceleration data was used for processing. 

As the SNR compares the level of the desired signal to the level of background noise, abrupt changes in movement with the resulting changes in acceleration will be registered by the SNR value. A low SNR value relates to “high” noise in the signal, or very abrupt changes in movement, which will make it difficult to determine the magnitude of the signal accurately. A high SNR value relates to “low” noise, or a smoother gait, which we relate to a steady walk. Therefore, the SNR of both acceleration and jerk can be regarded as a single measurement providing information on how the acceleration of the current user compares to a natural system variation. 

A common problem with data collection for gait analysis is the existence of noise in the acceleration signal. The extraneous stochastic component in the signal holds irrespective of the technique used, and it is present even in measurements with accelerometers attached to users while walking [6,7,8,15]. This requires the processing of time series data using techniques like moving average, where averages of different subsets of the full data set are calculated to smooth out short-term fluctuations and highlight longer-term trends or cycles, or employing low pass filters, which pass signals with a frequency lower than a selected cutoff frequency and attenuate signals with frequencies higher than the cutoff frequency, in order to interpret them. Following this, steps in the walk are individually identified and parameters related to gait are extracted. 

For this particular study an approach was implemented that processed the raw data of every step of the participants’ entire walk. After calculating the instantaneous acceleration and jerk of the ankle for each individual time point of the whole run, the SNR for the acceleration and the SNR for the jerk were calculated. Both SNR values that were calculated characterize all recorded steps, and combine both the phases and parameters of gait such as heel strike, stance, and push off in one single parameter.

Our approach is based on an assumption that the values of acceleration and jerk are adequate measures of stability. The justification is that these two parameters are being used in vehicles, elevators, and roller coaster design where it is necessary to limit the exposure of passengers to both the maximum force (acceleration) and maximum jerk since time is required to adjust muscle tension and get used to even modest stress changes [29]. It has been found that excessive jerking is associated with an uncomfortable ride, even at levels of acceleration that do not cause injury [30].

The next section describes the participants in the study and the protocol for data collection.

### 2.2. Participants 

A total of 10 participants were evaluated. Five disabled users and five non-disabled users, and one experienced temporary crutch user. The data presented in this paper is based on the following: (i) 15 videos taken of disabled users, who had been using crutches for many years because of their disabilities incurred by poliomyelitis, (ii) 46 videos taken of the five non-disabled users, with four using crutches sporadically for the needs of these measurements for periods ranging between one day to a few weeks, and (iii) 7 videos taken with an experienced temporary user who had been using crutches continuously for a period of six weeks due to multiple fractures of his leg. The summary of data is given in Table 1. The data have been collected over the period of 8 months, from June 2020 to January 2021.

The IRB approval for the study has been obtained from SUNY Korea on 7 August 2020, and it follows the Declaration of Helsinki and its set of ethical principles. 

The next section presents the results obtained from the analysis of video files from the 10 participants. 

## 3. Results

For the purpose of analysis, the participants are grouped in three categories: novice users, experienced temporary user, and disabled users. Table 2 presents the analysis of motion measurements for each group calculated from raw data. The left ankle acceleration of each user of the three groups was measured using the video analysis, with the left side chosen because it was weight bearing for the majority of users.

When a comparison is made, using the box and whisker chart, see Figure 3, between the accelerations of typical runs of each of the three groups studied in this work, the skewness in a typical dataset is visualized. The median line which cuts the box is not in the middle of it and the 25% quartile side, the bottom part, is very small, while the top part of the box, the 75% quartile, is quite larger as accelerations higher than the median show a wider range in that section. The upper and lower whiskers extent out to the extreme values of acceleration, while the outliers are not shown for clarity. 

At this stage of the analysis, the novice users’ group and the experienced temporary user show a very similar spread of results with the disabled users showing a much larger spread. 

However, once the results are further analyzed and processed a different picture emerges, one where differences in gait characteristics are not related to disability. Our results show that the acceleration SNR of the disabled users is the lowest at 2.1 m/s^2^, and the novice users at 3.9 m/s^2^, twice as that of the disabled users, and the experienced user recording the highest value of the three user groups at 8.8 m/s^2^. Although the walk of the disabled users was periodic, it was characterized by abrupt changes in movement due to their disability, which was verified by the actual videos. Novice crutch users showed a steady gait but lacked confidence and experience which resulted in intermediate acceleration SNR values when compared to the temporarily disabled user. 

A similar observation was made in the calculated jerk SNR. Jerk, taken from engineering system studies, measures the rapid changes in acceleration that the body experiences. In the previous biometric studies, it has been established that Parkinson’s disease patients showed larger jerk values, which authors used to measure the relative smoothness of postural sway and interpreted as a measure of dynamic instability [17]. Others used acceleration-based parameters to characterize the joint stability and jerk was defined as a potential stability measure, since anomalous jerk could be a symptom of inadequate body control [18].

However, keep in mind that in the present study, high values of SNR jerk represent a smoother walk. Our results show that the disabled users’ group had the lowest jerk SNR of 0.8 m/s^3^, the novice users an intermediate value at 2.4 m/s^3^ and the experienced user the highest value of jerk SNR at 3.5 m/s^3^.

The SNR of acceleration and jerk for all novice users runs over this 8-month period are shown in Figure 4, and for all experienced (disabled and one temporary user) participants are shown in Figure 5. The changes in acceleration displayed by the experienced users are accompanied by similar changes in jerk in practically every case, with the jerk being consistently lower than the acceleration. It suggests that the data for crutch users with more experience reflects a more confident walk, irrespective of the differences in gait. 

This consistent behavior observed in the disabled and temporary experienced users is not observed in the novice users. In some runs, the acceleration and jerk do not follow each other at all, while there are many runs where they match in values. This last point is interesting as it indicates a walk with highly abrupt changes in acceleration and movement. The values of acceleration and jerk appear to be far higher in most cases in the novice users’ group.

The experienced temporary user data is shown in Figure 6. It can be seen clearly that both the acceleration and jerk SNR follow each other consistently. The acceleration SNR shows a much higher value while the jerk SNR is consistently low. This is interpreted as a stable and confident walk with crutches where the user follows a clear path in a controlled manner. The difference in acceleration and jerk between the disabled users and experienced temporary user is related to the complex movement due to the disability level. 

## 4. Discussion

The proposed interpretation of the acceleration data allows us to differentiate between each of the three groups by the movement their legs perform in space. The disabled users have a low, i.e., noisy, acceleration SNR, as their legs perform a complex movement in three dimensions due to their disabilities. The novice users, who are not hindered by any disability, show an improved acceleration SNR twice that of the disabled users, while the experienced temporary user shows an acceleration SNR nearly four times almost that of the disabled users and twice that of the novice users. In effect, a low acceleration SNR value indicates a far more complex 3-dimensional movement of the leg of the user, which can be evidenced by the videos. A high acceleration SNR value points to a near 2-dimensional movement of the leg along a plane, something which the novice users perform to a certain extent while walking with crutches, and certainly, the experienced temporary user does confidently and steadily.

The interpretation of the jerk data allows for a similar observation regarding stability and movement control between each group. The low jerk SNR value measured for the disabled users, which is related to very high rates of change of acceleration, matches previous observations of a higher jerk for neurological patients [20]. An unstable walk registers as high rates of change of acceleration, which is associated with a more abrupt gait. Intermediate jerk values are encountered for the novice users, where jerk SNR values measured three times higher than those of the disabled users, an effect of a steadier movement of the leg on a 2-dimensional plane. The experienced temporary user has a jerk value for his walks which is four times that of the disabled users and 50% higher than that of the novice users. This value reflects the steady and controlled movement with crutches that appeared subjectively in the videos.

When the acceleration SNR data are studied together with the jerk SNR data, there appears to be a differentiation between groups depending on their experience with crutches. The acceleration SNR data for all novice users do not show a consistent trend or value over a long period of time, but it appears that for the majority of runs of different users, the acceleration and the jerk SNR have similar and comparable values. This observation does not hold true for the experienced temporary user and the disabled users where the acceleration SNR and the jerk SNR show the same trend without registering the same values. This is clearly confirmed with the experienced temporary user for all of his runs. Therefore, it appears that experience with using crutches allows for a much higher acceleration SNR value than a jerk SNR value. This observation was confirmed irrespective of the varying gait characteristics of all the experienced users, both disabled and temporary. Confidence and control of the user are demonstrated by relative steady and high acceleration values with low jerk SNR values regardless of their disability.

## 5. Conclusions

In this study we present preliminary results derived from the data of ten people walking with crutches. Three groups have been analyzed: novice users, experienced temporary user, and disabled users. The SNR metric was selected in this study because while walking with crutches, the movement includes a large component of noise or abrupt changes superimposed stochastically to the signal. The SNR metric takes that into account by incorporating standard deviation in the calculation. When the acceleration SNR data are studied together with the jerk SNR data, there appears to be a differentiation between groups depending on their experience with crutches.

The most interesting result is that the experienced temporary user data shows both the acceleration and jerk SNR follow its other consistently. This is interpreted as a stable and confident walk with crutches where the user follows a clear path in a controlled manner. The experienced temporary user has a jerk value for his walks which is 4 times that of the disabled users and 50% higher than that of the novice users. This value reflects the steady and controlled movement with crutches that appeared subjectively in the videos.

The approach is promising because it allows us to differentiate between confident users and ones with unstable movement. This differentiation is independent of the degree of disability. This approach could improve the assessment of crutch user stability. Improving assessment would allow correcting the gait of individuals and preventing unnecessary falls.

Future work will involve expanding the sample size of experienced users, focusing on those temporarily using assistive devices due to their smoother movements that are easier to study. Furthermore, future studies will seek to define a more precise function of acceleration to use in the calculations. That will involve analyzing individual walk cycles for more granularity in the data.

## Figures and Tables

**Figure 1 ijerph-18-03001-f001:**
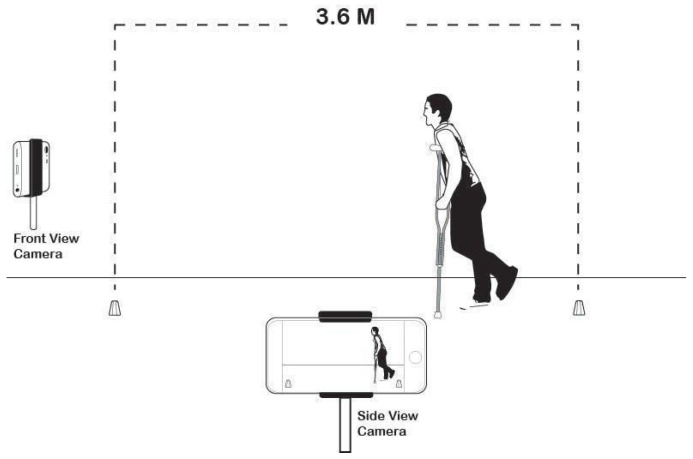
Walkway layout in the lab.

**Figure 2 ijerph-18-03001-f002:**
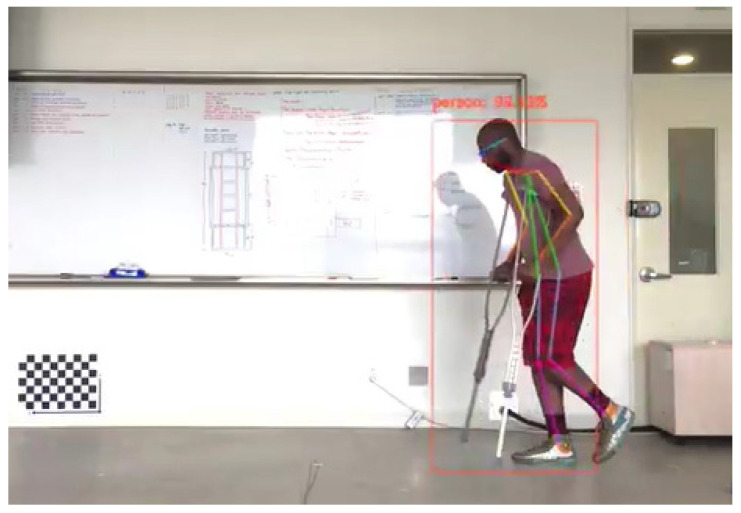
OpenPose output of one of the participants walking with crutches.

**Figure 3 ijerph-18-03001-f003:**
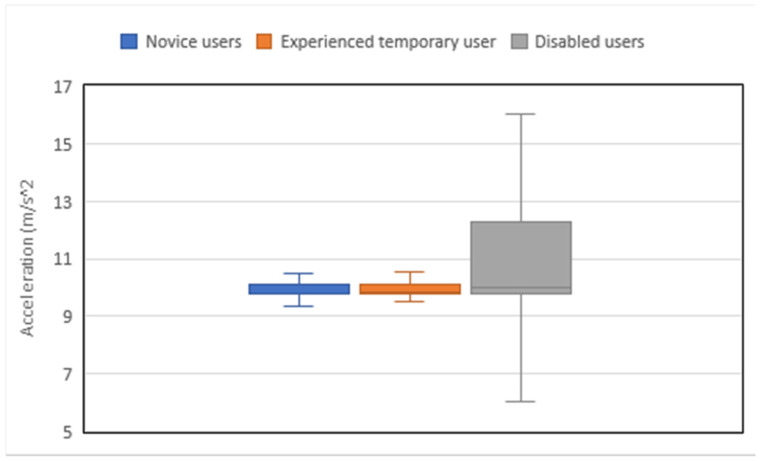
Typical user in each group.

**Figure 4 ijerph-18-03001-f004:**
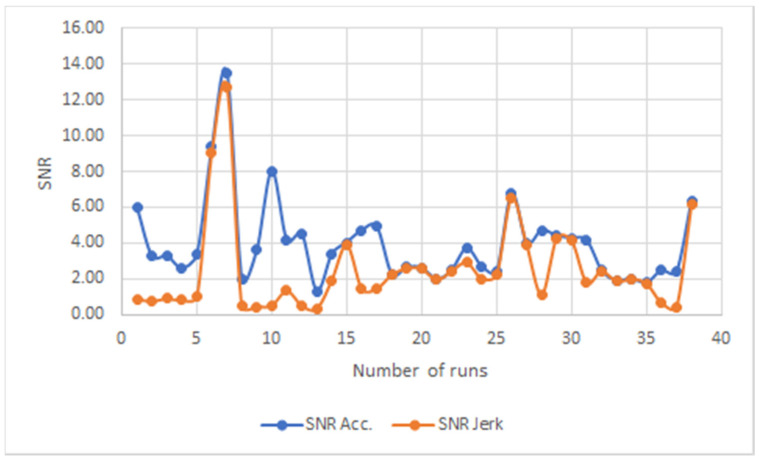
SNR for novice users.

**Figure 5 ijerph-18-03001-f005:**
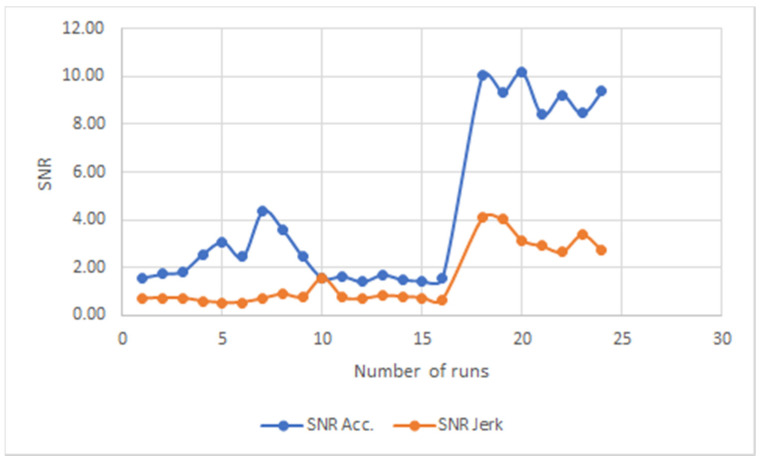
SNR for experienced users, both disabled and temporary.

**Figure 6 ijerph-18-03001-f006:**
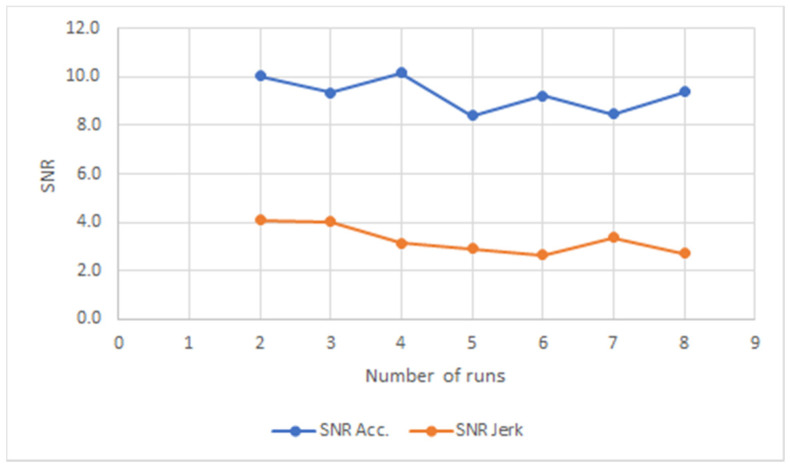
One experienced temporary user.

**Table 1 ijerph-18-03001-t001:** Number of video runs.

	No of Runs
Disabled Users	15
Novice Users	39
Experienced User	7

**Table 2 ijerph-18-03001-t002:** Analysis of motion measurement (average of all values (Avg.), maximum of all values (Max.), Standard Deviation of all values (Std. Dev), SNR of all values).

	Acceleration (m/s^2^)				Jerk (m/s^3^)			
	Avg.	Max.	Std. Dev.	SNR	Avg.	Max.	Std. Dev.	SNR
Disabled Users	12.4	53.1	6.7	2.1	103.3	777.2	144.8	0.8
Novice Users	10.7	36.6	3.6	3.9	208.7	909.9	112.4	2.4
Experienced temporary User	10.0	16.6	1.2	8.8	269.3	487.2	80.1	3.5

## Data Availability

The data from this study is open source and can be found here: https://www.mobilityaid.org/publications/gaitdata.

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
