# Peer review of "Assessing Stability of Crutch Users by Non-Contact Methods"

_ijerph, 2021, doi:10.3390/ijerph18063001_

Round 1
Reviewer 1 Report
The authors propose a non-contact method that can be used to assess the stability of people with assistive devices at a significantly lower cost than standard assessments that use expensive equipment. From video data, two values are calculated: Signal to Noise Ratio (SNR) of acceleration calculated as an inverse of the coefficient of variation of acceleration; and SNR of jerk (change in acceleration) to assess user’s stability while they walk with crutches
This paper presents work in an important field.
To the best of my knowledge, the paper is original and unpublished. The paper is well organized.
In the Introduction section, the paper should be devoted to give a comprehensive review of literature, papers on rehabilitation may also be included. It is recommended to analyze better the literature, more recent articles are available:
Evaluation of a Rehabilitation System for the Elderly in a Day Care Center, Information 2019, 10, 3.
Results section must be improved with comparisons with state of the art methods. In order to be published in a journal preliminary results are not enough, authors have to add more participants to prove the robustness of the method.
The paper needs to be revised and the English grammar has to be proofread by an English native speaker. References are not correctly reported in the text, reading the text has been difficult.
Author Response
1. In the Introduction section, the paper should be devoted to give a comprehensive review of literature, papers on rehabilitation may also be included. It is recommended to analyze better the literature, more recent articles are available: "Evaluation of a Rehabilitation System for the Elderly in a Day Care Center, Information 2019, 10, 3."
Thank you for pointing us towards literature in rehabilitation. We were able to find 4 more papers that are closely related to our study. The literature review has been updated and can be seen on page #3.
2. Results section must be improved with comparisons with state of the art methods. In order to be published in a journal preliminary results are not enough, authors have to add more participants to prove the robustness of the method.
We completely agree with you that the results would be more robust if we had more participants. As a matter of fact, our IRB allowed for 30 participants. However, we got interrupted by the current Covid restrictions on using outside participants and ended with only 11 participants. We are working with another university to use their already videotaped data to verify our methods but when we will get the data is uncertain. However, we feel that our results are interesting, meaningful, and significant to be published. In addition, many empirical studies have a sample of 10 and 15 people, so our sample is not significantly lower.
3. The paper needs to be revised and the English grammar has to be proofread by an English native speaker. References are not correctly reported in the text, reading the text has been difficult.
The resubmitted text has been proofread by a qualified English editor and references have been fixed.
Reviewer 2 Report
The manuscript is well written and well structured. There are several minor issues that need clarification, in particular in the section Materials and Methods.
- How did the authors ensure synchronization of recording on both cameras?
- Why were the features Acceleration and Jerk selected?
- Which vectors from OPenPose were used?
Author Response
- How did the authors ensure synchronization of recording on both cameras?
Thank you for pointing that issue out. We had explained it on page #4, the second paragraph "The two videos were manually synchronized ....."
- Why were the features Acceleration and Jerk selected?
We have explained that at the end of Section 2.1, second to last paragraph "The justification is that these two parameters..."
- Which vectors from OpenPose were used?
We have addressed that at the end of page #4, "The 2D pixel location values of the joints ..."
Reviewer 3 Report
It is a reasonable written paper including an innovative cheap method to do a gait analysis.
I have a few remarks to improve the paper.
- IRB in the sentence below table 1 is not defined; please include the full name of this Review Board.
- In Table 2 I would recommend to also include an outcome of the variation like the SD or Range in stead of only the mean value. So readers can see how much overlap there might be.
- In the figure 3 and 4 there is no unit mentioned of the SNR at the Y-axis
- Please include the definition of Jerk in the Introduction. I saw a short explanation in the Abstract but not in the Introduction.
- If this is the first time in literature you create/use the inverse of the cv of the SNR, you might include this fact by saying that you did not find this in literature and you have the following reasons to try this Inverse CV of SNR because that is innovation that the readers might want to know.
- It is for me a little unclear how you integrate both video signals; please add a few line about this essential aspect of your method.
Author Response
- IRB in the sentence below table 1 is not defined; please include the full name of this Review Board.
I am sorry but because of the anonymity of the first submission, we did not include the full name of the IRB. Now the sentence reads "The IRB approval for the study has been obtained from the [name of authors' university] university on August 7, 2020, and it follows the Declaration of Helsinki and its set of ethical principles." After the paper has been accepted we can add the name. Also, we have sent the IRB approval letter to the journal editor.
- In Table 2 I would recommend to also include an outcome of the variation like the SD or Range instead of only the mean value. So readers can see how much overlap there might be.
We have changed the table including the variables you recommended. Thank you.
- In the figure 3 and 4 there is no unit mentioned of the SNR at the Y-axis
We explain on page #3 "The use of SNR in our context is different and it has been calculated as the inverse of the coefficient of variation. Because of this the SNR that we calculate is dimensionless."
- Please include the definition of Jerk in the Introduction. I saw a short explanation in the Abstract but not in the Introduction.
At the end of page #2, we added a short description of a jerk as "... the rate of change of acceleration, ..." later on page #4, there is a more detailed one.
- If this is the first time in literature you create/use the inverse of the cv of the SNR, you might include this fact by saying that you did not find this in literature and you have the following reasons to try this Inverse CV of SNR because that is innovation that the readers might want to know.
We have included a statement with regard to that in the introduction.
- It is for me a little unclear how you integrate both video signals; please add a few line about this essential aspect of your method.
On page #4, we added a sentence "The 2D pixel location values of the joints were extracted from OpenPose and exported to the MATLAB software for 3D localization using stereo vision techniques. Furthermore, MATLAB extrapolated the 3D velocities, accelerations, and jerk from those positions." Also, the second paragraph in section 2.1 explains synchronization and integration in more detail. We hope it is more clear now.
Round 2
Reviewer 1 Report
The authors correctly revised the manuscript as suggested.